# Effects of Neurofeedback on the Working Memory of Children with Learning Disorders—An EEG Power-Spectrum Analysis

**DOI:** 10.3390/brainsci11070957

**Published:** 2021-07-20

**Authors:** Benito J. Martínez-Briones, Jorge Bosch-Bayard, Rolando J. Biscay-Lirio, Juan Silva-Pereyra, Lucero Albarrán-Cárdenas, Thalía Fernández

**Affiliations:** 1Departamento de Neurobiología Conductual y Cognitiva, Instituto de Neurobiología, Universidad Nacional Autónoma de México Campus Juriquilla, Querétaro QE 76230, Mexico; benjavik332@gmail.com (B.J.M.-B.); oldgandalf@gmail.com (J.B.-B.); albarranluc@hotmail.com (L.A.-C.); 2McGill Centre for Integrative Neuroscience (MCIN), Ludmer Centre for Neuroinformatics and Mental Health, Montreal Neurological Institute (MNI), McGill University, Montreal, QC H3A 2B4, Canada; 3Centro de Investigación en Matemáticas, Guanajuato GJ 36000, Mexico; rolando.biscay@cimat.mx; 4Facultad de Estudios Superiores Iztacala, Universidad Nacional Autónoma de México, Tlanepantla, Estado de México MX 54090, Mexico; jsilvapereyra@gmail.com

**Keywords:** neurofeedback, learning disorders, working memory, school-age children, EEG power spectrum, source localization

## Abstract

Learning disorders (LDs) are diagnosed in children impaired in the academic skills of reading, writing and/or mathematics. Children with LDs usually exhibit a slower resting-state electroencephalogram (EEG), corresponding to a neurodevelopmental lag. Frequently, children with LDs show working memory (WM) impairment, associated with an abnormal task-related EEG with overall slower EEG activity (more delta and theta power, and less gamma activity in posterior sites). These EEG patterns indicate inefficient neural resource management. Neurofeedback (NFB) treatments aimed at normalizing the resting-state EEG of LD children have shown improvements in cognitive-behavioral indices and diminished EEG abnormalities. Given the typical findings of WM impairment in children with LDs, we aimed to explore the effects of an NFB treatment on the WM of children with LDs by analyzing the WM-related EEG power spectrum. EEGs of 18 children (8–11 y.o.) with LDs were recorded, pre- and post-treatment, during performance of a Sternberg-type WM task. Thirty sessions of an NFB treatment (NFB-group, *n* = 10) or 30 sessions of a placebo-sham treatment (sham-group, *n* = 8) were administered. We analyzed the before and after treatment group differences for the behavioral performance and the WM-related EEG power spectrum. The NFB group showed faster response times in the WM task post-treatment. They also exhibited a decreased theta power and increased beta and gamma power at the frontal and posterior sites post-treatment. We explain these findings in terms of NFB improving the efficiency of neural resource management, maintenance of memory representations, and improved subvocal memory rehearsal.

## 1. Introduction

Learning disorders (LDs) are neurodevelopmental impairments, and they are found in 5%–20% of children and adolescents between 5 and 16 years old [1,2,3,4]. A child diagnosed with a specific LD has significant difficulties in learning the academic skills of reading, writing, or mathematics, with the development of these skills showing a significant lag for the age and schooling level [2]. A child with a combined deficiency in two or three of these skills belongs to a subtype of LD that was formerly known as LD not otherwise specified [5]. The co-occurrence of academic impairments appears in up to 80% of LD cases [6]. A specific learning disorder of reading is the most prevalent LD subtype, appearing alone or in combination with the other two specific disorders (writing or mathematics) in four out of five LD cases [2].

Coupled with the lagged development of academic skills, children with an LD usually endure a heterogeneous frame of cognitive impairments in processes such as phonological awareness, attentional control, processing speed, and working memory (WM) [6], the latter being the main source of this heterogeneity [7]. WM is the part of the memory in charge of the online processing of information in a type of limited-capacity mental workspace to achieve goal-directed actions [8]. This process is a commonly affected cognitive domain in children with an LD [9,10,11], and serves as an adequate predictor of current and future academic difficulties [12,13]. WM performance is more severely affected in children with an LD that co-occurs with other academic impairments [6]. A defective WM implies a diminished capacity for access, maintenance, and/or retrieval of information, usually of a phonological nature. School-age children require adequate WM functioning to properly develop their basic academic skills [12,14]. Children with an LD are also at an increased risk of suffering emotional disturbance in dealing with school issues [15,16]. An affective distress in LD often appears in the form of reduced self-esteem coupled with possible anxiety and depression problems that can be aggravated further into adolescence and adulthood [17].

Electroencephalography (EEG) is one of several approaches of neural data acquisition, the EEG signal reflecting the synchronous electrical activity of the brain [18]. The neural correlates of a child with LD have been identified with quantitative EEG analyses of a resting-state [19]. The resting-state EEG of LD children usually shows an abnormally slower EEG activity than age-matched children with typical development. The slower EEG activity of children with an LD is akin to that of younger healthy children, with more theta power in frontal regions and less alpha power in posterior (parietal and occipital) regions. This apparent lag in the brain functional development of children with LDs has led to the hypothesis of LDs as a developmental disorder with a delay in the EEG maturation that impairs their ability to keep up with a given grade at school [20,21,22].

Task-related EEG, recorded during the performance of WM tasks, has been examined using techniques such as event-related potentials (ERP) and power spectrum analysis. The Sternberg WM task [23] has been used because it isolates the different WM phases (i.e., encoding, maintenance, and retrieval). In an ERP study of poor readers vs. children with normal reading skills (control) who responded to a Sternberg WM task, it was found that poor readers had longer and larger P300 latencies at frontal sites for the retrieval phase [24]. These results show that a greater effort is required by children with LDs since the P300 amplitude is considered a marker of the amount of attentional resources required to perform a cognitive task [25]. Moreover, when the WM-related power spectrum of healthy children was compared with adults [26], the children showed more delta and theta power, and less alpha power. These EEG patterns were interpreted as compensatory mechanisms due to neural immaturity. These findings were supported by a study that compared children with LDs and healthy control children in a task-related power spectrum analysis of the maintenance phase of a WM task [27]. Children with LDs showed lower overall activity with more delta and theta power, and less gamma power at posterior brain sites, which is a pattern of activity considered indicative of inefficient neural resource management to achieve proper cognitive performance. In EEG studies during cognitive tasks, the delta activity has been implied with states of sustained concentration coupled with the inhibition of sensory information [26,28,29,30]. Higher task-related theta power is more pronounced in less apt individuals when performing more difficult tasks including situations that require a higher WM load and when focusing involves more effort [31,32,33,34,35,36]; hence, the task-related theta power is considered to be increasingly recruited according to the neural resources needed to properly perform a cognitive task. With regard to the gamma band, a sustained increase over posterior sites is involved with a role of memory maintenance and the binding of memory representations [37,38,39,40,41]. Thus, the previous findings point to greater recruitment of delta and theta power and less recruitment of the high-frequency gamma band in conditions demanding a higher WM load.

The main interventions used to treat LDs are special education classes and evidence-based programs of reading, writing, or mathematics [42,43,44]. Neurofeedback (NFB) treatment is a relevant therapeutic approach that has resulted from the EEG field of research. NFB is an operant conditioning training that aims to modify brain activity with therapeutic or performance-enhancing purposes [45,46,47]. NFB treatment still has experimental treatment status [48], with ongoing research of its effects in many disorders such as attention-deficit/hyperactivity disorder, anxiety disorders, epilepsy, and LDs [49,50,51]. The current research of NFB effects on LDs in children has shown that a protocol aimed at normalizing their altered EEG resting-state by reducing the theta/alpha ratio [51] is capable of boosting the cognitive-behavioral performance and improving resting-state EEG patterns, and the treatment effects are reported to last for at least two years [52]. These positive effects suggest a facilitation of the EEG maturation due to this NFB treatment. Two other works have also found that NFB benefits in children with LDs include improved spelling, increased EEG connectivity of the alpha-band with a measure of coherence [50], improved measures of reading and phonological awareness, and a normalization of EEG coherence measures [49].

Given that WM is frequently affected in children with LDs and that NFB treatments appear to boost cognitive-behavioral performance and regulate resting-state EEG, the goal of this work was to examine the effects of an NFB treatment (theta/alpha inhibition at lead with the most abnormal theta/alpha ratio) on the WM-related EEG power spectrum of children with LDs. Specifically, we aimed to analyze the WM-power spectrum of children with LDs during the maintenance phase of a Sternberg-type WM task, using a pre- and post-treatment comparison of an NFB treatment group and a sham-NFB treatment group (control). Our main hypothesis was that the NFB treatment would induce the normalization of the WM-related EEG power spectrum, by decreasing the EEG power in low frequencies within delta and theta bands and increasing high-frequencies within the gamma band, which would be in line with better cognitive performance [27].

## 2. Materials and Methods

### 2.1. Ethics

The Ethics Committee of the Instituto de Neurobiología of the Universidad Nacional Autónoma de México (UNAM) approved the experimental protocol on 1 July 2015 [INEU/SA/CB/146]. This protocol complies with the Ethical Principles for Medical Research Involving Human Subjects established by the Declaration of Helsinki [53]. An informed consent was signed by all the children who participated in the study and their parents.

### 2.2. Participants

Eighteen right-handed children (11 boys, 7 girls) aged 8 to 11 years diagnosed with an LD were selected from a larger sample of children referred by social workers from several elementary schools in Querétaro, México. All children fulfilled the following inclusion criteria: (1) a normal neurological and psychiatric assessment (except for the LD diagnostic requirements as stated below); (2) intelligence quotient (IQ) of at least 75 (Wechsler Intelligence Scale for Children 4th edition, (WISC-4) [54]), used to exclude children with an intellectual disability; (3) mother (or tutor, in her absence) with at least a completed elementary school education and a per capita income greater than 50 percent of the minimum wage; and (4) an abnormally high EEG theta/alpha ratio compared to a normative database [55]. The reason for considering the final inclusion criterion was that the EEG of children with an LD is characterized by having more theta and less alpha power than the EEG of children with typical development. We, therefore, calculated the z-value of the theta/alpha ratio (see Section 2.3) and selected children who presented with z-values greater than 1.645 (1-tailed distribution, *p* = 0.05) in at least one lead of their EEG spectra. The treatment (NFB or Sham) was delivered via the lead with the highest abnormal z-value.

The LD diagnosis was established based on the following three criteria: (a) poor academic achievement reported by teachers and parents; (b) percentiles at 16 or lower in the subscales of reading, writing, and/or mathematics of the Infant Neuropsychological Scale for Children [56]; and (c) the diagnosis of the LD was performed by a psychologist according to the DSM-5 criteria for LDs [2]. While several of the children failed on different items in the attentional evaluation of the DSM-5, as is common in this disorder [57,58], they did not meet the DSM-5 criteria of ADHD [2]. The frequencies of academic impairments found in our LD sample were as follows: 9 children were impaired in all three domains (reading, writing, and mathematics); 3 children were impaired in reading and writing; 2 children were impaired in reading and mathematics; 2 children were impaired in writing and mathematics; and 2 children were impaired in mathematics. We acknowledge a lack of homogeneity in our sample of LD children; however, WM deficits have been found for all LD subtypes [59,60], which is in line with our aims.

The children were randomly assigned to one of two groups: The NFB group (*n* = 10, 3 females) received an NFB treatment that reinforced the reduction of the theta/alpha ratio, and the Sham group (*n* = 8, 4 females) received a sham NFB treatment.

Independent non-parametric Mann–Whitney U tests were performed for the following variables: Full-scale IQ, WM index, and the theta/alpha ratio of the most abnormal EEG lead. As shown in Table 1, the pre-treatment descriptive characteristics reveal a proper random assignment of the children to each group, with no statistical differences in age, IQ, female/male ratio, or resting-state theta/alpha ratio before the treatment.

### 2.3. The z-Value of the Theta/Alpha Ratio

A resting-state EEG was recorded under an eyes-closed condition for sample selection purposes while the child was seated in a dimly lit, faradized, and soundproofed room, in the 19 leads of the 10–20 International System (ElectroCap™ Inc, Eaton, OH, USA), referenced to linked earlobes (A1A2). For this, we used a Medicid™ IV system (Neuronic Mexicana, SA, Mexico City, Mexico) and Track Walker™ v2.0 software. The amplifier bandwidth was set to between 0.5 and 50 Hz. All electrode impedances were a maximum of 10 kΩ, and the signal was amplified with a gain of 20,000. EEG data were sampled with a frequency of 200 Hz and edited offline. On average, 24 artifact-free segments of 2.56 s were used for analysis. 

To obtain the theta(θ)/alpha(α) ratio (θ/α), first, the absolute power (AP) of the broad-band model was calculated in the frequency domain, and the θ/α was obtained as the ratio of the AP(θ) and the AP(α) for each lead. Here, we used the theta and alpha frequency bands in their traditional definition: theta comprises the frequencies 3.5–7.5 Hz and alpha comprises the frequencies 7.8–12.5 Hz, with a frequency resolution of 0.39 Hz. For each lead κ, the equation of its θ/α can be defined as:(1)[θα]κ=APκ(θ)APκ(α)
(2)$$\mathrm{where}AP_{\kappa }(\theta )=\frac{1}{N_{\theta }}\sum_{f=1}^{N_{\theta }}Spec_{\kappa }(\theta (f)),\;\theta =[3.5:0.39:7.5]$$
and $Spec_{\kappa }(\omega )$ is the traditional EEG spectrum for the lead κ, at frequency *ω*, obtained via the fast Fourier transform (FFT) applied to the recorded EEG segments:(3)$$Spec_{\kappa }(\omega )=\frac{1}{2\pi *N_{p}*N_{seg}}\sum_{j=1}^{N_{seg}}V_{\kappa j}^{}(\omega )\cdot V_{\kappa j}^{}{\left(\omega \right)}^{*}$$
(4)whereVκj(ω)=∑t=0Np−1Vκj(t)⋅e−i2πtωNp

$V_{\kappa j}^{}(\omega )$ is the FFT coefficient for the lead κ, at frequency *ω*, in segment *j*, *N_p_* is the number of time points in each EEG segment, and *N_seg_* is the number of segments. The method to calculate $Spec_{\kappa }(\omega )$ is described in detail in [61,62].

A similar definition was used for AP(α).

To calculate the z value of the theta/alpha ratio (z[θ/α]), we obtained the populational age-dependent means (μ(age)) and standard deviations (σ) for the eyes-closed resting-state EEG for each lead used in our study. This was performed by calculating the θ/α index in each lead for all subjects of the Cuban normative database [55] and calculating 2nd-order polynomial age-dependent regressions of those θ/α indices for the normative data to obtain μ(age) and σ. This procedure was the same as described in [61,62].

### 2.4. Neurofeedback and Sham Treatments

The NFB treatment was applied using a neurofeedback program adapted by Díaz-Comas [51] for the Medicid IV recording system. In this program, a 1280 ms segment is automatically selected and the theta/alpha ratio is calculated as described in Section 2.3. This ratio is compared to the threshold value previously established by the therapist, and if the theta/alpha ratio is lower than the threshold value, a tone of 500 Hz at 60 dB (positive reinforcer) is emitted. In the continuous EEG recording, the 1280 ms segment is moved 20 ms forward each time. This process is repeated until EEG recording finishes using overlapped segments. The criterion for establishing this threshold the first time was using the subject’s value in their resting-state EEG recorded in the sample selection phase (Section 2.3), but this was adjusted by trial and error until the tone was delivered approximately 70% of the time. Later (every 3 min), it was verified whether the percentage of time remained between 60–80% of the 3-min period, and if so, the threshold was not modified further. If the tone appeared for more than 80% of the time (this situation is most common), the therapist changed the threshold value to a lower value. Likewise, if the tone appeared for less than 60% of the time, the threshold was increased.

The Sham treatment was identical to the NFB treatment, except that the reward was delivered randomly, non-contingent with the EEG activity of the child. The goal of a sham-NFB treatment is that subjects “feel” that they are receiving the real-NFB treatment, i.e., they should receive the same reward used for real-NFB, but this reward must be non-contingent with their brain activity; therefore, the cue for this is in the reward used, not from where it comes. There are several ways to apply this: one is using the reward produced by another participant’s brain activity, another is using the reward randomly produced. Both ways imply that the reward is unrelated with the participant’s own brain activity. In other studies, it has been shown that some of the participants who receive the sham treatment “reported finding the feedback confusing and ineffective” [63]; no child in our Sham group reported something akin to this. In our study, none of the participants knew which condition they were in, nor did they know there were experimental and control conditions.

Each subject received 30 training sessions three times a week over a period of 10–12 weeks, and the duration of each session was 30 min. At the beginning of each session, the child was told that they would receive a candy at the end of the session depending on their performance. For motivational purposes, a learning-curve plot was updated each session, showing the last successful theta/alpha ratio in that session.

All of the children were examined with the WISC-4 pre- and post-treatment, and the EEG during a Sternberg-type WM task was recorded. 

### 2.5. Working Memory Task

The WM task used in this work was a modified version of the Sternberg memory task [23], a classic task used to assess each phase of the WM process (encoding, maintenance, and retrieval). A verbal version of this task was employed, since children with LDs show a more consistent deficit in the phonological loop subsystem of the Baddeley’s WM model [6,8].

The WM task (Figure 1) consisted of two conditions (Low-Load and High-Load) presented in 180 trials, with 90 trials per condition appearing randomly. At each trial, four digits appeared simultaneously on the screen after a warning signal (asterisk). In the Low-Load condition, all digits were the same; in the High-Load condition, the digits were different and did not appear in ascending or descending order, nor were they purely even or odd numbers. The participants were instructed to memorize the numbers after the set appeared, and a single digit (probe stimulus) was presented. Participants had to press one button (match response) if the digit was included in that trial and another button if not (non-match response). To perform the power spectrum analysis, segments of 800 ms were selected in the WM maintenance phase in trials with correct answers. Stimuli were presented with the software MindTracer [64] and synchronized with the EEG data acquisition system. This WM task was administered twice for both groups, that is, before the treatment (NFB or Sham) and two months after the treatment. 

### 2.6. EEG Recording and Data Analysis of the WM Task

Before and after the treatment, the administration of the WM task was coupled with an EEG of similar specifications to the resting-state condition. All the children were seated in the faradized dim-lit and soundproofed room. The task-related EEG was recorded during the task performance (with eyes open) using the Medicid IV and Track Walker^TM^ v2.0 data systems, from 19 leads of the 10–20 system referenced to the linked earlobes (A1–A2). The amplifier bandwidth was set between 0.1 and 50 Hz. The signal was amplified with a gain of 20,000 and electrode impedances were at most 5 kΩ. The sampling frequency of the EEG data was 200 Hz.

The power spectra were calculated using windows of 800 ms corresponding to the WM maintenance phase from each trial with correct responses. For each condition, there were up to 90 trials recorded to guarantee a sufficiently high number of EEG epochs for the analysis. 

To calculate the cross-spectral matrices for the analysis, we used an average of 24 quasi-stationary and artifact-free EEG epochs, with a minimum set of 19 epochs per condition. Under these conditions, the EEG power spectrum was smooth and the cross-spectral matrix was positively defined, a requirement for applying the inverse solution procedure to estimate the distribution of the primary currents at the sources [61].

EEG preprocessing was performed offline by an expert neurophysiologist, who visually selected only quasi-stationary and artifact-free epochs before the probe-stimuli onset. No automatic rejection algorithm was employed. 

To project the EEG signals on the sources, we used the s-Loreta technique [65], which transferred our data from 19 leads to a high-resolution volumetric grid of 3244 sources. 

Projecting the EEG scalp voltage to the EEG generators in the brain gray matter is a necessary step to overcome the volume conduction problem and the reference electrode effect at the scalp, which mixes and distorts the real neurophysiological information due to the high mixing of signals [65,66,67,68].

However, projecting the EEG over the sources did not completely resolve the mixing of the signals due to the low-resolution nature of the inverse methods, especially linear methods such as s-Loreta. The amount of mixing at the sources was in relation to the resolution matrix of the specific method. For linear methods, Biscay et al. [68] found that only a small number of sources can be estimated independently for a given number of EEG sensors (specifically the number of electrodes minus 1). They proposed an algorithm that unmixes the source signals for that small number of sources that works under mild assumptions. In particular, the selected sources (in general they can also be regions of interest (ROIs)) must be separated by a distance greater than the resolution matrix of the inverse method. In this study, we adhered to that methodology.

For s-Loreta estimation, the EEG epochs were re-referenced from the linked earlobes to the Average Reference. This step guaranteed that the estimated primary current at the sources was reference-free [65].

To select the specific sources (or ROIs) to be used in the analysis, we applied a data-driven approach where the ROIs were selected based on the intrinsic variability of the data. For this purpose we developed a variant of the eigenvector centrality mapping technique (ECM) [69]. The ECM technique is based on the calculation of the first principal component of the time signals over all voxels. This procedure produces a global connectivity index for the corresponding voxels. The voxel is considered to be more connected in the brain when the connectivity index is higher. The algorithm that we implemented to calculate the ECM was an optimized version of the power method that reduces memory usage and decreases CPU intensity, which allowed us to obtain the first principal component for all of the subjects simultaneously. In this way, the global connectivity index represents a populational index of connectivity.

Based on the global connectivity index, we selected 18 ROIs (the number of scalp sensors minus 1). However, to have a symmetric representation of the sources in the two hemispheres, in those cases where the corresponding index was not selected by the procedure in the contralateral hemisphere, we included that ROI by hand. In Figure 2, we show the 18 selected ROIs using our approach, which was also published by Martínez-Briones et al. [27]. To illustrate how distant these ROIs were from the recording sites, we plotted the cortex regions nearest to the positions of the sensors at the scalp in blue. Note that many of the relevant areas detected by our algorithm are far from the sources immediately below the scalp sensors.

The next step was to apply the unmixing algorithm elaborated by Biscay et al. [68] to the time signals of the 18 selected ROIs. Then, the epochs of unmixed signals of the 18 ROIs (each containing 160 time points) for all subjects in each condition were transformed to the frequency domain by means of the fast Fourier transform (FFT) and the log spectra for the 18 ROIs were calculated in the frequency range from 1.25 to 50 Hz (40 frequency bins every 1.25 Hz) for every ROI, subject, and condition.

To compare the EEG spectra of the two groups (NFB and Sham) before and after the neurofeedback intervention, we performed statistical analysis of the power spectrum using this narrow band model of 1.25 Hz frequency resolution up to 50 Hz. In the Results and Discussion sections, we refer to our findings using the classic frequency bands: delta (δ) = 1–4 Hz, theta (θ) = 4–8 Hz, alpha (α) = 8–12 Hz, beta (β) = 12–30 Hz, and gamma (γ) = 30–50 Hz (the upper extreme of the interval was never reached to avoid overlapping). The gamma band is usually reported up to 100 Hz; however, we report changes up to 50 Hz due to our hardware limitations, which is considered a lower-gamma band.

We used a linear mixed-effects model (LME) [70] to test the EEG spectra at each frequency to compare the two groups before and after treatment.

The permutations technique [71] was applied to correct the thresholds of significance of our results given the high number of comparisons. These levels are shown in each figure as two horizontal lines, indicating the upper and lower significance thresholds at *p* = 0.05.

For behavioral analysis of the Sternberg WM task (correct responses and response times), non-parametric independent Wilcoxon signed-rank tests were performed.

## 3. Results

According to the comparison of the main characteristics of both groups before treatment (see Table 1), the NFB and Sham groups did not differ in age, gender, IQ, or theta/alpha ratio. The changes in IQ measures and theta/alpha ratio were of interest for comparisons between the groups, but they did not statistically differ. The statistical significance within-group of the differences between theta/alpha ratio before and after treatment was assessed by a multivariate nonparametric permutation test [72] for dependent variables. Significant reduction was observed for both groups (NFB group: t = 2.11, *p* = 0.03, Cohen’s d = 1.15; Sham group: t = 2.44, *p* = 0.01, Cohen’s d = 2.2). No significant changes in IQ were observed for any group.

### 3.1. Behavioral Results of the WM Task

The behavioral results of the WM task are shown in terms of a percentage of correct responses and the response times of the two conditions (Low-Load and High-Load). There were fewer correct responses and slower response times in the High-Load condition than in the Low-Load condition. This pattern of differences appears for the two groups both before and after treatment, suggesting that the High-Load condition task was indeed more difficult at this behavioral level.

We assessed within-group differences for the percentage of correct responses (Figure 3) and response times (Figure 4) by comparing the pre- and post-treatment results for each group separately. In these comparisons, we did not find statistical differences in the percentage of correct responses for either group, a finding that could point to insufficient sensitivity of the WM task to detect possible improvements in performance at this behavioral level. However, for the response times, the NFB group did show a faster response time for the High-Load condition after the NFB treatment (W = 27, *p* = 0.01). Thus, the NFB treatment seems to modify an index of good performance in terms of an improved velocity of WM retrieval. 

### 3.2. WM-Related Power Spectrum Results

For the EEG power spectrum analysis, we focused on the WM High-Load condition, given that our within-group High-Load vs. Low-Load comparisons did not show statistical differences at this power spectrum level.

We performed a comparison between treatment groups (NFB vs. Sham) after subtracting the before from the after-treatment data for each separate group, resulting in an ‘after minus before’ new variable for the contrast between the groups (Figure 5). This comparison was made to assess the differences between the groups in terms of the effect produced by each of the treatments, since it is known that placebo procedures can produce some positive [70,71] and even negative effects in subjects [73]. Differences were observed in frequencies within the theta, beta, and gamma bands. The specific results that this analysis yielded for the theta band included differences between groups localized in the frontal areas (except for the right lateral orbitofrontal gyrus) and posterior areas, such as the left superior parietal lobule and the right occipital pole. Differences between groups in terms of the effect produced by the corresponding treatment were found in frequencies within the upper-beta and gamma bands at all bilateral frontal and parietal areas (including a higher gamma power at the left occipital pole).

Figure 6 and Figure 7, shown below, represent the post hoc analyses of the previous comparison. Figure 6 shows the power spectrum within-group differences for the NFB group by comparing the pre- and post-treatment conditions, while Figure 7 shows the same comparison for the Sham group. The NFB group after NFB treatment showed a decreased delta power in the right parietal areas (right superior parietal lobule and angular gyrus), and a decreased theta power in all bilateral frontal areas (except for the right lateral orbitofrontal gyrus) and the bilateral parietal areas (superior parietal lobules and angular gyri). In all other ROIs we found an increased upper beta (except for the bilateral temporal gyri and the left occipital pole) and an overall increased gamma power (except for the bilateral middle temporal gyri). The Sham group showed a somewhat distinct pattern of differences after the placebo treatment with an increase in theta power in the bilateral frontal area (except for the bilateral orbitofrontal gyri), left superior parietal lobule, and right occipital pole. The Sham group also showed an increased beta power at the right occipital pole, but mainly exhibited a decrease in beta power in all frontal areas (except for the bilateral orbitofrontal gyri) and bilateral parietal areas (bilateral superior parietal lobules and angular gyri), along with lower gamma power in the frontal areas (except for the left-middle, medial and inferior frontal gyri) and in the left occipital pole. Therefore, the NFB group showed a selective modulation of the power at low-frequency bands in the form of a decrease in frequencies within the delta band at parietal sites and theta band at frontal and parietal areas, and an increase in high-frequencies (beta and gamma) in frontal and posterior regions after NFB treatment. In contrast, the Sham group mainly exhibited a power increase in frequencies within the theta band in the frontal and posterior areas and a decrease in power of high-frequencies (within beta and gamma bands) at frontal and posterior sites after the Sham-NFB treatment.

## 4. Discussion

In this study, we explored the effects of an NFB treatment on the WM processing of children with LDs and an excessive theta/alpha ratio in their resting-state EEG. To this end, we compared the behavior and the WM-related EEG power spectrum between a group of children with LDs who received an NFB treatment and another group of children with LDs who were given a Sham-NFB treatment.

In a pre-treatment descriptive comparison of the groups, we did not find statistical differences for the main variables of age, gender, IQ (including a WM index provided by the WISC-4 test), theta/alpha ratio, or WM behavioral performance (measured by correct response and response time). Thus, our random assignment of children with LDs successfully ensured that groups were comparable in the WM post-treatment behavioral and power spectrum results. Our primary outcomes of interest were those regarding our selected Sternberg’s memory task both at the behavioral and the EEG level. However, it must be noted that neither group showed a post-treatment improvement in IQ, WM index, or the theta/alpha ratio. The main reason for measuring IQ in this study was to satisfy the criterion for the diagnosis that establishes that learning difficulties are not better accounted for by an intellectual disability [2]. In general, IQ is a rigid measure with a high rate of failure to be improved by therapy or programs with performance-enhancing aims [74,75,76]; thus, our negative finding was likely to occur. Schooling has been found to improve the IQ of subjects with a typical development at 1–5 points for every additional year of education [77]. Since the IQ improvement does not usually happen in children with LDs, this only adds to the importance of finding out more about possible treatments for people with LDs who both struggle at school and whose usual WM impairments also contribute to their academic challenges [14].

Both groups showed a theta/alpha ratio reduction in the lead considered for the treatment. In the NFB group this is an expected and desirable result since the reduction is an index of the learning produced by the operant conditioning involved in the NFB treatment. In the Sham group some reduction would be anticipated too due to expectation [78], the placebo effect [79], and meta-cognitive processes [80], frequently included as non-controlled variables in this treatment, with the capacity to induce changes in the EEG. On the other hand, because the theta/alpha ratio is reported in z values, it is not influenced by maturation due to increased age. 

With regard to the behavioral results of the WM task, besides the expected Low-Load vs. High-Load within-group differences in terms of the correct response rate and response time [27], our additional statistical comparisons yielded a main difference in the NFB group. In the pre- and post-treatment comparison of each separate group, the NFB group showed a faster response time for the High-Load condition after the NFB treatment. Hence, the NFB treatment appeared to improve the speed of WM retrieval in children with an LD. A good WM performance is required for proper academic achievement, and a better response time in a task that involves memorizing digits is a noteworthy finding. 

The WM-related power spectrum analysis was realized not in the sensor space but for 18 source ROIs. An adapted eigenvector centrality mapping (ECM) technique was used as a data-driven procedure to select the ROIs. This yielded a global index of connectivity for each voxel that allowed a more robust algorithm for ROI selection. This data-driven procedure is a valuable ROI selection approach that avoids the assumption of arbitrary or uninformed criteria such as choosing the sources closer to the leads, or supposed prior knowledge of brain structure or function, such as an alleged WM network that could not apply to children with LDs with insufficiently mapped task-related neural correlates, or who possibly employ a different strategy to solve a task. By contrast, our ROIs broadly underlie the sample variance as active sites present in the children during the maintenance phase of the WM performance. A main result from this approach was of many ROIs being selected in prefrontal areas, and no ROIs selected around the central sulci, i.e., near the Cz, C3, and C4 leads. This finding agrees with other task-related EEG studies that did not identify a contribution of central regions during cognitive performance, while mainly frontal and posterior regions have been shown to be involved in WM functioning [35,81,82,83].

Regarding the EEG power spectrum analysis during the maintenance phase of the WM task, the comparison between groups in terms of the effect produced by each of the treatments yielded differences in theta, upper-beta, and gamma power at frontal and posterior areas. Therefore, in the post hoc analysis we focused on the pre- and post-treatment changes in these bands and regions.

We examined each group taken separately in a within-group before vs. after treatment comparison. Our original hypothesis was that NFB induces a tendency to normalize the WM-related power spectrum by diminishing the delta and theta power while increasing gamma activity. We found these predicted patterns in the NFB group, showing specific low-frequency decreases of parietal delta and frontal-posterior theta power, coupled with high-frequency increases of beta and gamma power at frontal-posterior sites), suggesting that the NFB treatment induces a tendency to normalize the function underlying WM processing. Conversely, in the Sham group, an increase in theta power at frontal areas (and left superior parietal lobule) coupled with frontal-posterior beta and gamma power decreases was observed, indicating that the WM in these children moved away from normality, accentuating their brain dysfunction. Other studies have reported possible adverse effects of Sham-NFB interventions, such as the occurrence of learned helplessness [73], inducing higher levels of restlessness or anxiety by the nature of the noncontingent random reward that fails to be predicted by the child. It could be interesting to explore the relationship between these adverse effects and the changes observed in the Sham group. 

A greater theta power during mental processing occurs in more cognitively demanding situations related with an increased recruitment of neural resources [27,34,36]. Gamma activity has been attributed to a role of memory maintenance and the binding of memory representations [40,41], while beta activity is related both to a subvocal rehearsal during the retention of items [83] and the preparation of motor responses [28]. The main finding in our NFB group after treatment that the global theta power decreased could be explained as an EEG tendency to normality, with the theta changes signifying an improved efficiency in the management of neural resources, a development otherwise lacking in the Sham group. Regarding the gamma changes, we cannot detect the binding of memory representations because our equipment does not allow us to record beyond 50 Hz and our power spectrum analysis was performed over EEG segments taken in the maintenance phase. We assume that the increased gamma power of the NFB group indicates that memory maintenance was improved due to the NFB treatment, a finding that could also be an EEG substrate of the improved speed of WM retrieval for this same group of children. On the other hand, the increased beta activity in the NFB group was directly related to WM tasks as an index of subvocal memory rehearsal [83]. Although the beta increase has been related to motor performance [28], the motor component is absent during this phase of the task. Moreover, such beta increase has been observed as a result of anxiety-reducing therapeutic interventions [84], i.e., forms of meditation including mindfulness training, which could share some positive effects with biofeedback and NFB treatments [85,86]. Yet, there is conflicting evidence of beta power changes after meditation programs, with some studies reporting increases and others decreases in beta power [87,88]. Thus, the beta power changes after NFB treatment could signify a specific improvement in the WM subvocal rehearsal during the retention of digits mixed with nonspecific effects in the level of relaxation for our children with LDs. 

Our group has previously found positive results of the NFB treatment in children with LDs over a two-year follow-up in measures of behavioral performance and resting-state EEG [52]. We intend to follow through with this verification step for the WM results reported here in a future study.

## 5. Limitations

The main limitation of this study is the small size of both samples. Thus, we used non-parametric statistics for being distribution-free. However, this limitation consequently brought a reduction in the statistical power of our results, leaving this work as an exploratory study.

It could be assumed that it is a weakness of the study finding no differences between groups in the reduction of the theta/alpha ratio in the lead considered for training. However, there may be several explanations for this. First, as a consequence of the small sample sizes it may be insufficient to detect small changes, especially if there was high within-group variability. Second, in previous studies of our research group using this NFB protocol, we always found a reduction (in average) in the theta/alpha ratio [51,88,89], but this finding was not common to all individuals who received the NFB treatment [51,88]. This could be a consequence of a NFB mechanism that we previously hypothesized [51,89]: NFB treatment could primarily modify the functioning of subcortical structures, which would not necessarily be reflected in cortical postsynaptic activity; thus, it would be unlikely to observe EEG changes since 97% of the recorded EEG activity originates in the cortex [67,90]. Nevertheless, these functional changes of deep structures could later modify the EEG through the modulation of thalamic-cortical circuits [52,91]. However, there is no reason to suppose that these subsequent changes in the cortical activity will produce EEG modifications in the specific lead taken as the reference for treatment. In addition to the evidence provided by Fernández et al. [51] on LDs, Lubar et al. [92] on ADHD, and Sterman and Egner [93] on epilepsy, there is indirect evidence that by regulating a range of frequencies in the EEG for a single lead, the final changes are observed in other frequencies and different regions, which points to a certain non-specificity of frequency and location effects in NFB.

Lastly, because of the small-size samples, we could not perform special analyses to test if neurofeedback success correlates with any outcome.

## 6. Conclusions

This is the first study to investigate the effects of an NFB treatment on WM measures at the behavioral and the EEG power spectrum levels in children with LDs. We obtained promising positive results including improved response times post-treatment; a decreased theta power and increased beta and gamma power in the frontal and posterior areas. We found that the power spectrum patterns of a diminished theta power indicate an improved efficiency of neural resource management and the boost in the gamma band as revealing improved maintenance of memory representations due to NFB, coupled with the increased beta activity as an index of improved subvocal rehearsal. 

## Figures and Tables

**Figure 1 brainsci-11-00957-f001:**
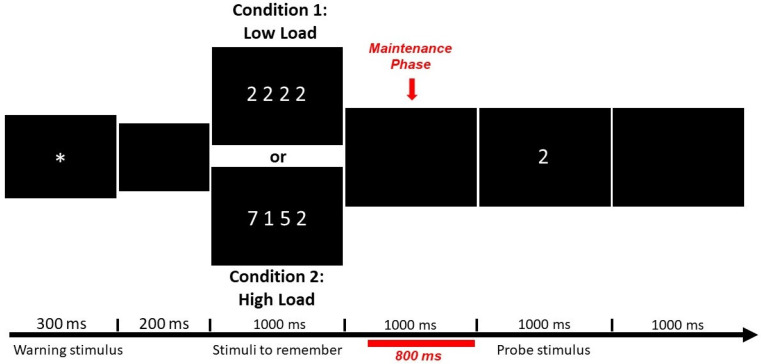
Representation of a single trial showing an example of Low-Load and High-Load conditions. In this case, the single digit (‘probe stimulus’) was included previously in the set ‘stimuli to remember’ from both conditions, and the subject had to press the button of the ‘match response’. The segment in red corresponds to the WM maintenance phase, the section selected for the power spectrum analysis. The total trial duration was 4500 msec. Modified from Martínez-Briones et al. [27].

**Figure 2 brainsci-11-00957-f002:**
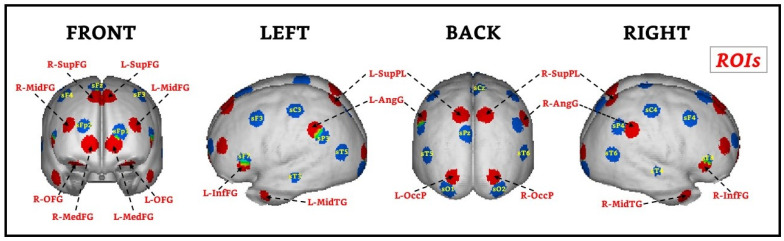
ROIs selected by the populational ECM. The sources closer to the 19 leads are shown in blue, and the 18 ROIs are shown in red. L-OFG: Left lateral orbitofrontal gyrus; R-OFG: Right lateral orbitofrontal gyrus; L-MedFG: Left medial frontal gyrus; R-MedFG: Right medial frontal gyrus; L-InfFG: Left inferior frontal gyrus; R-InfFG: Right inferior frontal gyrus; L-MidFG: Left middle frontal gyrus; R-MidFG: Right middle frontal gyrus; L-SupFG: Left superior frontal gyrus; R-SupFG: Right superior frontal gyrus; L-MidTG: Left middle temporal gyrus; R-MidTG: Right middle temporal gyrus; L-SupPL: Left superior parietal lobule; R-SupPL: Right superior parietal lobule; L-AngG: Left angular gyrus; R-AngG: Right angular gyrus; L-OccP: Left occipital pole; R-OccP: Right occipital pole. Modified from Martínez-Briones et al. [27].

**Figure 3 brainsci-11-00957-f003:**
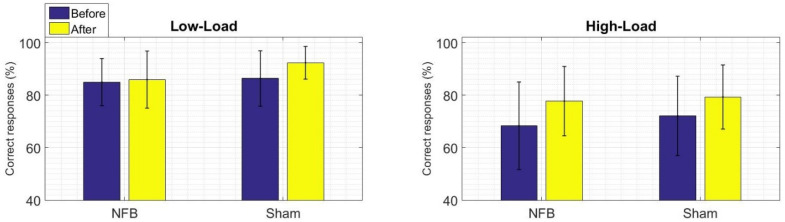
Within-group behavioral results for the percentage of correct responses for the WM task (the left panel shows the Low-Load condition, the right panel shows the High-Load condition). Mean values of the percentage of correct responses before treatment appear in blue and after treatment appear in yellow. There were no statistical differences (Wilcoxon signed-rank test) for the before vs. after treatment comparisons within groups.

**Figure 4 brainsci-11-00957-f004:**
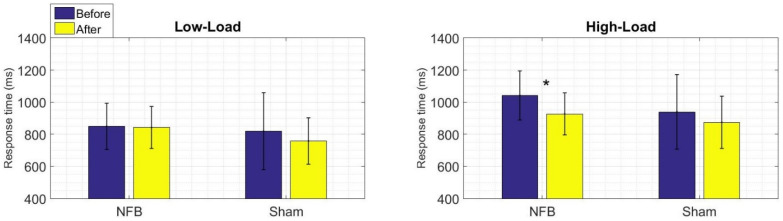
Within-group behavioral results for the response times to the WM task (the left panel shows the Low-Load condition, the right panel shows the High-Load condition). Mean values of response time before and after treatment are shown in blue and yellow, respectively. Asterisks indicate statistically significant differences (Wilcoxon signed-rank test) in the before vs. after treatment comparison of the NFB group for the High-Load condition (W = 27, * *p* = 0.01).

**Figure 5 brainsci-11-00957-f005:**
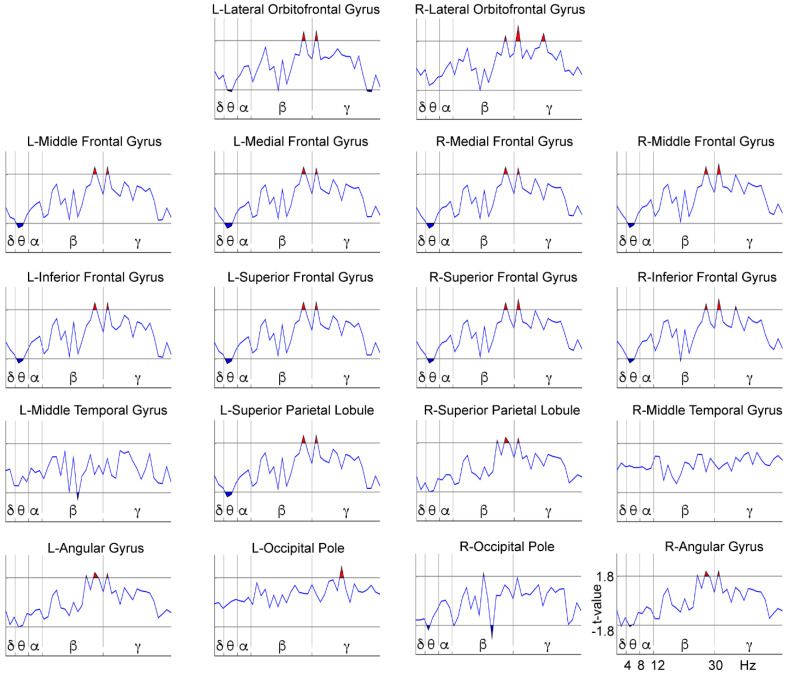
EEG power spectrum differences between the groups (NFB vs. Sham) for the High-Load after subtracting the before-treatment from the after-treatment condition (yielding an ‘after minus before’ condition of group comparisons). The X-axis represents the frequencies (1.25–50 Hz) separated by vertical lines to show the classic frequency bands: delta (δ) = 1–4 Hz, theta (θ) = 4–8 Hz, alpha (α) = 8–12 Hz, beta (β) = 12–30 Hz, and gamma (γ) = 30–50 Hz (open upper intervals). The Y-axis represents the t-values of the LME procedure. The red patches above the horizontal lines indicate a higher power for the NFB group than for the Sham group (*p* < 0.05, randomization-corrected). The blue patches below the horizontal lines indicate a higher power for the Sham group (*p* < 0.05, randomization-corrected). L = Left; R = Right.

**Figure 6 brainsci-11-00957-f006:**
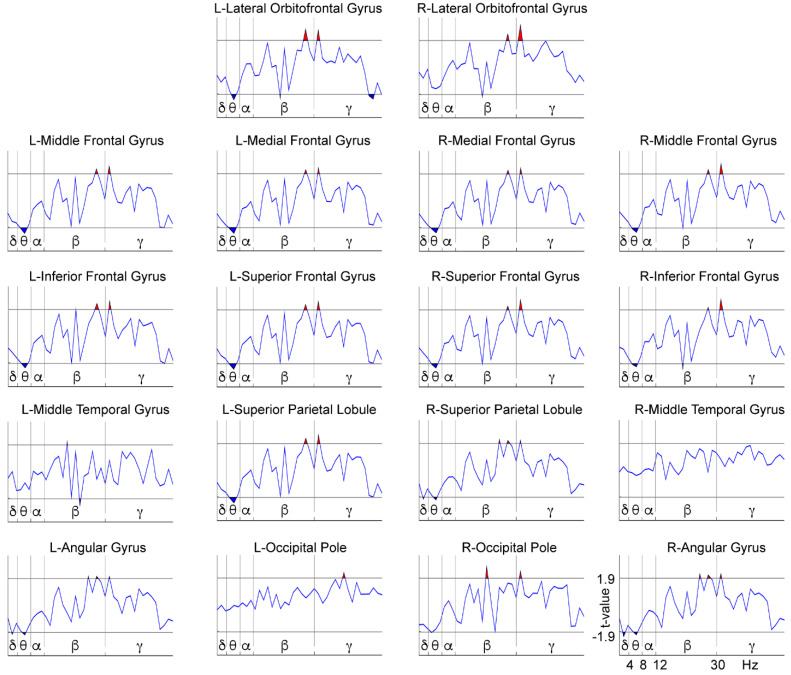
Differences between before and after treatment for the EEG power spectra in the WM High-Load condition within the NFB group. The X-axis represents the frequencies (1.25–50 Hz), which are separated by vertical lines according to the classic frequency bands: delta (δ) = 1–4 Hz, theta (θ) = 4–8 Hz, alpha (α) = 8–12 Hz, beta (β) = 12–30 Hz, and gamma (γ) = 30–50 Hz (open upper intervals). The Y-axis represents the t-values of the LME procedure. The red patches (above the horizontal lines) indicate a higher power for the after-treatment condition than for the before-treatment condition (*p* < 0.05, randomization-corrected). The blue patches (below the horizontal lines) indicate a higher power for the before-treatment condition (*p* < 0.05, randomization-corrected). L = Left; R = Right.

**Figure 7 brainsci-11-00957-f007:**
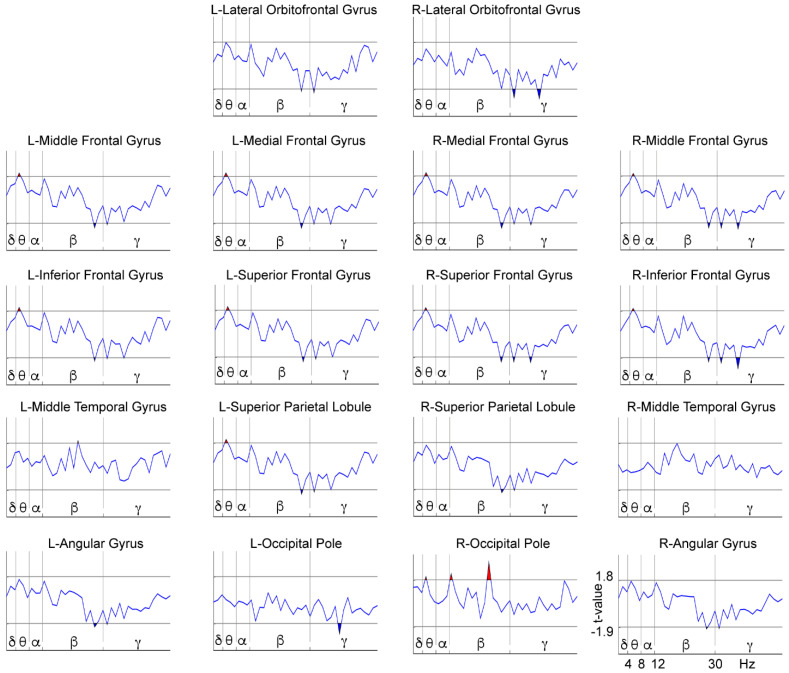
Differences between before and after treatment for the EEG power spectra in the WM High-Load condition within the Sham group. The X-axis represents the frequencies (1.25–50 Hz), which are separated by vertical lines according to the classic frequency bands: delta (δ) = 1–4 Hz, theta (θ) = 4–8 Hz, alpha (α) = 8–12 Hz, beta (β) = 12–30 Hz, and gamma (γ) = 30–50 Hz (open upper intervals). The Y-axis represents the t-values of the LME procedure. The red patches (above the horizontal lines) indicate a higher power for the after-treatment condition than for the before-treatment condition (*p* < 0.05, randomization-corrected). The blue patches (below the horizontal lines) indicate a higher power for the before-treatment condition (*p* < 0.05, randomization-corrected). L = Left; R = Right.

**Table 1 brainsci-11-00957-t001:** Sample composition before treatment.

	NFB Group *n* = 10		Sham Group *n* = 8		Statistical Differences between Groups
	Mean	sd	Mean	sd	
Age	10.4	1.0	10.1	0.8	U = 26, *p* = 0.37
WISC-4 test:					
Full scale IQ	90.1	12.4	89	8.5	U = 30, *p* = 0.6
WM index	85	11.7	94.5	13.7	U = 23, *p* = 0.23
Female/male ratio	3/7		4/4		*p* = 0.63, Fisher’s exact test
z Theta/alpha ratio *	2.6	0.8	2.2	0.6	U = 22, *p* = 0.34

* The z-value of the theta/alpha ratio refers to the lead with the most abnormal EEG theta/alpha ratio.

## Data Availability

The data used in this study are part of a bigger project under development at the Laboratory of Psychophysiology of the Instituto de Neurobiología (UNAM), and will be publicly shared as a whole when the study is concluded. However, for interested researchers, the data will be available under reasonable direct request to the corresponding author Thalia Fernandez <thaliafh@yahoo.com.mx>. The code used in this paper will also be shared under direct request to Jorge Bosch-Bayard <jorge.boschbayard@mcgill.ca>.

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
