# Peer review of "Effects of Neurofeedback on the Working Memory of Children with Learning Disorders—An EEG Power-Spectrum Analysis"

_brainsci, 2021, doi:10.3390/brainsci11070957_

Round 1
Reviewer 1 Report
Undoubtedly, learning disorders are diagnosed in children impaired in the academic skills of reading, writing and/or mathematics. I share the opinion that children with LD usually show a slower resting-state electroencephalogram EEG, with EEG patterns corresponding to a neurodevelopmental lag. In this artcile Authors analyzed the before-after treatment group differences for the behavioral performance and the WM-related power-spectrum.
My comments to the article are as follows:
- As part of the Introduction, I propose to provide a broader background in the description of methods of acquiring signals from the human brain. For example, I suggest referring to: Data Acquisition Methods for Human Brain Activity, Analysis and classification of eeg signals for brain-computer interfaces, Book Series: Studies in Computational Intelligence, Springer from 2020.
- I propose to extend the argument on the basis of which a given group of children was selected for research.
- I propose to consider shortening the descriptions under the figures in favor of a detailed reference to them in the text. Especially in the case of Figures: 5, 6 and 7.
- Please describe the equation contained in line 179/180 with index / number and correct its notation. Please refer to the index / number in the text.
- The results of the research in the form of data recording in Figs. 5, 6, 7 are hardly visible. I propose to enlarge them and place them as Appendix.
- The article lacks Conlusions, please add them. Some of the information from Discussion can be carried over to Conclusions.
Author Response
Thank you very much for your suggestions. We think they have been very useful to improve our article.
R1.1.- As part of the Introduction, I propose to provide a broader background in the description of methods of acquiring signals from the human brain. For example, I suggest referring to: Data Acquisition Methods for Human Brain Activity, Analysis and classification of EEG signals for brain-computer interfaces, Book Series: Studies in Computational Intelligence, Springer from 2020.
A: Thank you for your suggestion. This new reference was added in the Introduction (lines 65-66): Electroencephalography (EEG) is one of several approaches of neural data acquisition, the EEG signal reflecting the synchronous electrical activity of the brain [18]
R1.2.- I propose to extend the argument on the basis of which a given group of children was selected for research.
A: We recognize a lack of homogeneity in the cognitive deficit domains of our sample; therefore, we added a justification for the composition of our sample based on the objectives of this study (i.e., the effects of NFB on maintenance of WM). The specific cognitive deficit (in reading, writing, or mathematics) is not so important because all LD subtypes have in common the presence of WM deficits [59, 60]. (151-157): The frequency of academic impairments found in our LD sample were as follows: 9 children were impaired in all three domains (reading, writing, and mathematics); 3 children were impaired in reading and writing; 2 children were impaired in reading and mathematics; 2 children were impaired in writing and mathematics; and 2 children were impaired in mathematics. We acknowledge a lack of homogeneity in our sample of LD children; however, WM deficits have been found for all LD subtypes [59,60], which is in line with our aims.
R1.3.- I propose to consider shortening the descriptions under the figures in favor of a detailed reference to them in the text. Especially in the case of Figures: 5, 6 and 7.
A: We shortened the description of the figures by including the full names of the ROIs in the graphs instead of in the captions.
R1.4.- Please describe the equation contained in line 179/180 with index / number and correct its notation. Please refer to the index / number in the text.
A: Instead the equation for a specific age, we used a 2nd-order polynomial age-dependent regression. The procedure is described in the new section 2.3 of the Method (lines 171-196).
R1.5.- The results of the research in the form of data recording in Figs. 5, 6, 7 are hardly visible. I propose to enlarge them and place them as Appendix.
A: Figures 5, 6, and 7 were corrected and enlarged. We think that it is not necessary to include them in an Appendix. Order was changed and new figures are as follows: old-5=new-6, old-6=new-7, old-7=new-5.
R1.6.- The article lacks Conclusions, please add them. Some of the information from Discussion can be carried over to Conclusions.
A: We added the section “5. Conclusion” (lines 561-569):
- Conclusion
This is the first study to investigate the effects of an NFB treatment on WM measures at the behavioral and the EEG power spectrum levels in children with LDs. We obtained promising positive results including improved response times post-treatment; a decreased theta power and increased beta and gamma power in the frontal and posterior areas. We found that the power spectrum patterns of a diminished theta power indicate an improved efficiency of neural resource management and the boost in the gamma band as revealing improved maintenance of memory representations due to NFB, coupled with the increased beta activity as an index of improved subvocal rehearsal.
Reviewer 2 Report
Title: “Effects of Neurofeedback in the Working Memory of Children with Learning Disorders: An EEG Power-Spectrum Analysis”
In this work the authors aimed at exploring the effects of a NFB treatment in the WM of children with LD, by analyzing the WM-related EEG power-spectrum. They recruited 18 children with LD (8-10 years old). They performed a Sternberg-type WM-task synchronized with an EEG of 19 leads (10-20 system) twice n pre-post treatment conditions. They went through either 30 sessions of a NFB treatment (NFB-group, n= 10); or through 30 sessions of a placebo-sham treatment (Sham-group, n= 8). In addition, the authors analyzed the before-after treatment group differences for the behavioral performance and the WM-related power-spectrum. The NFB group showed faster response times in the WM-task post-treatment. They also showed an increased gamma power at posterior sites and a decreased beta power. The authors explained these findings in terms of NFB improving the maintenance of memory representations coupled with a reduction of anxiety.
General comment: This work seems to be interesting. The organization of the overall manuscript seems to be adequate. Nevertheless, this manuscript still need to be reworked to improve its quality and impact.
Some detailed comments:
Materials and methods:
*) The authors should better organize this section. They should provide all details, write in a clear way equations and better describe the statistical methodologies used in this work.
Lines: "The log value (theta AP/alpha AP) was computed, and z-values for this logarithm 178
were calculated using the equation: 179
Z= [log (thetaAP/alphaAP) - / 180
where and are the mean value and the standard deviation, respectively, of a 181
normative sample of the same age as the subject (Szava et al., 1994; Valdés et al., 1990). 182"
*) The authors should provide equations in a clear way and provide them of a suitable number reference
*) it is not totally clear what statistical methods have been used by the authors to prove the statistically relevant difference of specimens. Could the authors better explain ? Are the chosen methods suitable for a small amount of subjects ?
Results
Lines: “Figure 3. Within-groups behavioral results of the percentage of correct responses for the WM task 369 (the left panel shows the Low-Load condition, the right panel shows the High-Load condition). 370 Mean values of the percentage of correct responses before treatment appear in blue and after 371 treatment appear in yellow. There were no statistical differences for the before vs. after treatment 372 comparisons within groups taken separately”
*) The statistical treatment is not clear. Please explain better
Figure 4. Within-groups behavioral results of the response times for the WM task (the left panel 375
shows the Low-Load condition, the right panel shows the High-Load condition). Mean values of 376
response time before treatment appear in blue, and after treatment appear in yellow. The asterisk 377
indicates statistically significant differences in the before vs. after treatment comparison of the NFB 378
group for the High-load condition (t= 2.56, *p<0.05). 37
*) Also here the statistical treatment is not explicitly provided.
Lines: “Figure 5. Power-spectrum differences within the NFB group in a before vs. after treatment com- 401
parison of the WM High-Load. The X-axis represents the frequencies (1.25-50 Hz), separated by 402
vertical lines to the classic frequency bands: delta (δ)= 1-4 Hz, theta (θ)= 4-8 Hz, alpha (α)= 8-12 Hz, 403
beta (β)= 12-30 Hz, and gamma (γ)= 30-50 Hz (open upper intervals). The Y-axis represents the t-values 404
of the LME procedure. The red patches (above the horizontal lines) indicate a higher power for the 405
after-treatment condition than for the before-treatment condition (p*<0.05, randomiza- 406
tion-corrected). The blue patches (below the horizontal lines) indicate a higher power for the be- 407
fore-treatment condition (p*<0.05, randomization-corrected). LatFOGL/LatFOGR: Left/Right lateral 408
orbitofrontal gyrus; MedFGL/MedFGR: Left/Right medial frontal area; InfFGL/InfFGR: Left/Right 409
inferior frontal gyrus; MidFGL/MidFGR: Left/Right medium frontal gyrus; SupFGL/SupFGR: 410
Left/Right superior frontal gyrus; MidLTGL/MidLTGR: Left/Right medial temporal gyrus; Sup- 411
PLL/SupPLR: Left/Right superior parietal area; AngGL/AngGR: Left angular gyrus; OccPL/OccPR: 412
Left/Right occipital pole.”
*) Perhaps the authors could improve this caption making it more understandable.
Lines: “Figure 6. Power-spectrum differences within the Sham group in a before vs. after treatment com- 415
parison of the WM High-Load. The X-axis represents the frequencies (1.25-50 Hz), separated by 416
vertical lines to the classic frequency bands: delta (δ)= 1-4 Hz, theta (θ)= 4-8 Hz, alpha (α)= 8-12 Hz, 417
beta (β)= 12-30 Hz, and gamma (γ)= 30-50 Hz (open upper intervals). The Y-axis represents the t-values 418
of the LME procedure. The red patches (above the horizontal lines) indicate a higher power for the 419
after-treatment condition than for the before-treatment condition (p*<0.05, randomiza- 420”
*) See the previous comment.
Lines: “Figure 7. Power-spectrum differences between the groups (NFB vs. Sham) for the High-Load after 442
subtracting the before-treatment from the after-treatment condition (yielding an ‘after minus be- 443
fore’ condition of group comparisons). The X-axis represents the frequencies (1.25-50 Hz), separated 444
by vertical lines to the classic frequency bands: delta (δ)= 1-4 Hz, theta (θ)= 4-8 Hz, alpha (α)= 8-12 445
Hz, beta (β)= 12-30 Hz, and gamma (γ)= 30-50 Hz (open upper intervals). The Y-axis represents the 446
t-values of the LME procedure. The red patches (above the horizontal lines) indicate a higher 447
power for the NFB group than for the Sham group (p*<0.05, randomization-corrected). The blue 448
patches (below the horizontal lines) indicate a higher power for the Sham group (p*<0.05, ran- 449
domization-corrected). LatFOGL/LatFOGR: Left/Right lateral orbitofrontal gyrus; MedFGL/MedFGR: 450
Left/Right medial frontal area; InfFGL/InfFGR: Left/Right inferior frontal gyrus; MidFGL/MidFGR: 451
Left/Right medium frontal gyrus; SupFGL/SupFGR: Left/Right superior frontal gyrus; Mid- 452
LTGL/MidLTGR: Left/Right medial temporal gyrus; SupPLL/SupPLR: Left/Right superior parietal 453
area; AngGL/AngGR: Left angular gyrus; OccPL/OccPR: Left/Right occipital pole. 45”
*) See the previous comments.
Lines: “To conclude, this is the first study of the effects of a NFB treatment in WM measures 566
(at the behavioral and the EEG power spectrum levels), showing promising positive re- 567
sults in variables such as improved response times post-treatment and an increased 568
gamma power at the parietal areas coupled with a decreased beta power by the NFB 569
treatment. We explicate these power spectrum patterns of a boost in the gamma band as 570
revealing improved maintenance of memory representations due to NFB; coupled with 571
the decreased beta band as an index of reduced anxiety. Our group has previously found 572
positive results of NFB in LD children over a two-year follow-up [51] and we also aim to 573
follow through with this verification step for our WM results.”
*) Starting from figures 3 and 4, the authors should demonstrate their claim.
Author Response
We are very grateful for your suggestions. We think they have been very useful to improve our article
Materials and methods:
R2.1.- The authors should better organize this section. They should provide all details, write in a clear way equations and better describe the statistical methodologies used in this work.
A: We reorganized the Method with the aim of providing better clarity overall and to diminish the percentage of self-plagiarism.
R2.2.- Lines: "The log value (theta AP/alpha AP) was computed, and z-values for this logarithm were calculated using the equation:
Z= [log (thetaAP/alphaAP) - / 180
are the mean value and the standard deviation, respectively, of as and mwhere normative sample of the same age as the subject (Szava et al., 1994; Valdés et al., 1990). 182"
*) The authors should provide equations in a clear way and provide them of a suitable number reference
A: Instead the equation for a specific age, we used a 2nd-order polynomial age-dependent regression, and added a number reference. The procedure is described in the new section 2.3 of the Method (lines 171-196).
R2.3.- *) it is not totally clear what statistical methods have been used by the authors to prove the statistically relevant difference of specimens. Could the authors better explain ? Are the chosen methods suitable for a small amount of subjects ?
A: We clarified our statistical method and justified its usage to analyze our data. Now non-parametric tests were used. We believe the statistical methods are appropriate for small samples since they do not assume any distribution of the data.
Results
R2.4.- Lines 368-373: “Figure 3. Within-groups behavioral results of the percentage of correct responses for the WM task (the left panel shows the Low-Load condition, the right panel shows the High-Load condition). Mean values of the percentage of correct responses before treatment appear in blue and after treatment appear in yellow. There were no statistical differences for the before vs. after treatment comparisons within groups taken separately”
*) The statistical treatment is not clear. Please explain better.
A: We clarified our statistical method to analyze our data (lines 317-321, 333-334).
R2.5.- Lines 375-379: Figure 4. Within-groups behavioral results of the response times for the WM task (the left panel shows the Low-Load condition, the right panel shows the High-Load condition). Mean values of response time before treatment appear in blue, and after treatment appear in yellow. The asterisk indicates statistically significant differences in the before vs. after treatment comparison of the NFB group for the High-load condition (t= 2.56, *p<0.05).
*) Also here the statistical treatment is not explicitly provided.
A: We clarified our statistical method to analyze our data (lines 317-321, 333-334).
R2.6.- Lines 401-412: “Figure 5. Power-spectrum differences within the NFB group in a before vs. after treatment comparison of the WM High-Load. The X-axis represents the frequencies (1.25-50 Hz), separated by vertical lines to the classic frequency bands: delta (δ)= 1-4 Hz, theta (θ)= 4-8 Hz, alpha (α)= 8-12 Hz, beta (β)= 12-30 Hz, and gamma (γ)= 30-50 Hz (open upper intervals). The Y-axis represents the t-values of the LME procedure. The red patches (above the horizontal lines) indicate a higher power for the after-treatment condition than for the before-treatment condition (p*<0.05, randomization-corrected). The blue patches (below the horizontal lines) indicate a higher power for the before-treatment condition (p*<0.05, randomization-corrected). LatFOGL/LatFOGR: Left/Right lateral orbitofrontal gyrus; MedFGL/MedFGR: Left/Right medial frontal area; InfFGL/InfFGR: Left/Right inferior frontal gyrus; MidFGL/MidFGR: Left/Right medium frontal gyrus; SupFGL/SupFGR: Left/Right superior frontal gyrus; MidLTGL/MidLTGR: Left/Right medial temporal gyrus; Sup- PLL/SupPLR: Left/Right superior parietal area; AngGL/AngGR: Left angular gyrus; OccPL/OccPR: Left/Right occipital pole.”
*) Perhaps the authors could improve this caption making it more understandable.
A: We shortened the description of the figures by including the full names of the ROIs in the graphs instead of in the captions.
R2.7.- Lines 414-420: “Figure 6. Power-spectrum differences within the Sham group in a before vs. after treatment comparison of the WM High-Load. The X-axis represents the frequencies (1.25-50 Hz), separated by vertical lines to the classic frequency bands: delta (δ)= 1-4 Hz, theta (θ)= 4-8 Hz, alpha (α)= 8-12 Hz, beta (β)= 12-30 Hz, and gamma (γ)= 30-50 Hz (open upper intervals). The Y-axis represents the t-values of the LME procedure. The red patches (above the horizontal lines) indicate a higher power for the after-treatment condition than for the before-treatment condition (p*<0.05, randomiza- 420”
*) See the previous comment.
A: We shortened the description of the figures by including the full names of the ROIs in the graphs instead of in the captions.
R2.8.- Lines 442-: “Figure 7. Power-spectrum differences between the groups (NFB vs. Sham) for the High-Load after subtracting the before-treatment from the after-treatment condition (yielding an ‘after minus before’ condition of group comparisons). The X-axis represents the frequencies (1.25-50 Hz), separated by vertical lines to the classic frequency bands: delta (δ)= 1-4 Hz, theta (θ)= 4-8 Hz, alpha (α)= 8-12 Hz, beta (β)= 12-30 Hz, and gamma (γ)= 30-50 Hz (open upper intervals). The Y-axis represents the t-values of the LME procedure. The red patches (above the horizontal lines) indicate a higher power for the NFB group than for the Sham group (p*<0.05, randomization-corrected). The blue patches (below the horizontal lines) indicate a higher power for the Sham group (p*<0.05, randomization-corrected). LatFOGL/LatFOGR: Left/Right lateral orbitofrontal gyrus; MedFGL/MedFGR: Left/Right medial frontal area; InfFGL/InfFGR: Left/Right inferior frontal gyrus; MidFGL/MidFGR: Left/Right medium frontal gyrus; SupFGL/SupFGR: Left/Right superior frontal gyrus; MidLTGL/MidLTGR: Left/Right medial temporal gyrus; SupPLL/SupPLR: Left/Right superior parietal area; AngGL/AngGR: Left angular gyrus; OccPL/OccPR: Left/Right occipital pole. 45”
*) See the previous comments.
A: We shortened the description of the figures by including the full names of the ROIs in the graphs instead of in the captions.
R2.9.- Lines 565-573: “To conclude, this is the first study of the effects of a NFB treatment in WM measures (at the behavioral and the EEG power spectrum levels), showing promising positive results in variables such as improved response times post-treatment and an increased gamma power at the parietal areas coupled with a decreased beta power by the NFB treatment. We explicate these power spectrum patterns of a boost in the gamma band as revealing improved maintenance of memory representations due to NFB; coupled with the decreased beta band as an index of reduced anxiety. Our group has previously found positive results of NFB in LD children over a two-year follow-up [51] and we also aim to follow through with this verification step for our WM results.”
*) Starting from figures 3 and 4, the authors should demonstrate their claim.
A: We clarified our claims in the Discussion (lines 537-555):
The main finding in our NFB group after treatment that the global theta power decreased could be explained as an EEG tendency to normality, with the theta changes signifying an improved efficiency in the management of neural resources, a development otherwise lacking in the Sham group. Regarding the gamma changes, we cannot detect the binding of memory representations because our equipment does not allow us to record beyond 50 Hz and our power spectrum analysis was performed over EEG segments taken in the maintenance phase. We assume that the increased gamma power of the NFB group indicates that memory maintenance was improved due to the NFB treatment, a finding that could also be an EEG substrate of the improved speed of WM retrieval for this same group of children. On the other hand, the increased beta activity in the NFB group was directly related to WM tasks as an index of subvocal memory rehearsal [87]. Although the beta increase has been related to motor performance [28], the motor component is absent during this phase of the task. Moreover, such beta increase has been observed as a result of anxiety-reducing therapeutic interventions [88], i.e., forms of meditation including mindfulness training, which could share some positive effects with biofeedback and NFB treatments [87,88]. Yet, there is conflicting evidence of beta power changes after meditation programs, with some studies reporting increases and others decreases in beta power [86,89]. Thus, the beta power changes after NFB treatment could signify a specific improvement in the WM subvocal rehearsal during the retention of digits mixed with nonspecific effects in the level of relaxation for our children with LDs.

Round 2
Reviewer 1 Report
Dear Authors,
Thank you for answering my questions.
Thank you for the changes made. Currently, I have no comments to the study.
I recommend the article for publication.
Author Response
Dear Reviewer 1,
The authors are very grateful for your comments, which contributed to improve our article. However, we are attaching a new version to add an explanation on how to obtain the power spectrum using FFT.
Sincerely,
Thalía

Reviewer 2 Report
Title: "Effects of Neurofeedback in the Working Memory of Children with Learning Disorders: An EEG Power-Spectrum Analysis"
General comment: The authors revised their work according to the previous suggestions.
However, lines 467-473 where the Equations (1) and (2) are presented is still not too clear. Please clarify in a better way.
Indeed, it seems that interested readers are only referred to previous literature without any other information.
Author Response
Dear Reviewer 2,
The authors are very grateful for your comments, which contributed to significantly improve our article. In this new version we are adding an explanation on how the power spectrum is obtained using the FFT; however, we refer the interested reader to the cited bibliography [61, 62] if he/she wishes to delve into it.
We enclosed the "CLEAN" version of Round 1 with the tracked addition.
Sincerely,
Thalía
